# Diffusion Language Models are Provably Optimal Parallel Samplers

**Haozhe Jiang  Nika Haghtalab  Lijie Chen**
University of California, Berkeley
{ericjiang,nika,lijiechen}@berkeley.edu

## Abstract

Diffusion language models (DLMs) have emerged as a promising alternative to autoregressive models for faster inference via parallel token generation. We provide a rigorous foundation for this advantage by formalizing a model of parallel sampling and showing that DLMs augmented with polynomial-length chain-of-thought (CoT) can simulate any parallel sampling algorithm using an optimal number of sequential steps. Consequently, whenever a target distribution can be generated using a small number of sequential steps, a DLM can be used to generate the distribution using the same number of optimal sequential steps. However, without the ability to modify previously revealed tokens, DLMs with CoT can still incur large intermediate footprints. We prove that enabling remasking (converting unmasked tokens to masks or revision (converting unmasked tokens to other unmasked tokens) together with CoT further allows DLMs to simulate any parallel sampling algorithm with optimal space complexity. We further justify the advantage of revision by establishing a strict expressivity gap: DLMs with revision or remasking are strictly more powerful than those without. Our results not only provide a theoretical justification for the promise of DLMs as the most efficient sampler, but also advocate for why revisions should be enabled in DLMs.

## 1 Introduction

Diffusion language models (DLMs, Austin et al. (2021); Nie et al. (2025); Song et al. (2025)) have recently emerged as a promising alternative to autoregressive (AR) generation, fueled by their potential to deliver faster generation and more efficient inference. In contrast to AR models—which, despite running on massively parallel hardware, must still generate text sequentially, token by token—DLMs decode by iteratively denoising masked sequences, allowing many positions to be unmasked in parallel. This fundamental architectural difference offers the promise of significantly reducing generation and inference latency, and has led to a surge of interest in DLMs in a world increasingly attuned to the value of inference-time scaling Song et al. (2025); Khanna et al. (2025).

Yet DLMs remain in their infancy compared to the highly optimized AR models, and it is far from clear when, or if, their promised advantages will translate into principled gains. This makes theory especially important for evaluating the genuine potential of DLMs: unlike AR models, whose capabilities under chain-of-thought (CoT; Wei et al. (2022)) reasoning are well understood, the basic limits of DLMs, such as *how much sequential computation they truly save, and what space and memory such parallelism requires*, remain largely unexplored.

In this work, we provide such a theoretical foundation for studying DLMs by formalizing them through the lens of circuit complexity, using circuit *depth* and *width* as abstractions for the time and space resources required to compute a function in parallel. As we describe in more detail below, our results establish the expressive power of DLMs as the most efficient parallel sampler.

Our first result (Theorem 3.1) shows that when equipped with sufficiently long CoT, DLMs can simulate any sampling procedure with the minimal number of sequential computational steps. Specifically, any distribution that can be realized by a circuit of depth $d$ can be generated by a DLM in $d$ decoding rounds, thereby matching the minimum number of sequential steps required by the target computation. This stands in stark contrast to AR decoding. In particular, while it is known that autoregressive models equipped with CoT can simulate any Boolean circuit (Li et al., 2024), the

number of sequential steps in this case (which in the cases of AR models equals the length of the CoT) grows linearly with circuit *size*, i.e., the total number of nodes, rather than with depth.

Beyond the number of steps required to generate a sample, we also analyze the working memory footprint (Theorem 3.2) of DLMs and show that it depends critically on design choices unique to this family of models—most notably on inference-time mechanisms such as *remasking* (Nie et al., 2025) and *revision* Song et al. (2025). Remasking is an inference-time mechanism that allows already unmasked tokens to be re-masked (noised) and resampled (denoised). Revision, on the other hand, permits unmasked tokens to be changed directly to other tokens. We show that when DLMs are augmented with CoT and either remasking or revision, they can simulate any circuit using memory that scales only linearly with the circuit's width.

Finally, we demonstrate via the problem of uniformly sampling from strings with zero parity that remasking or revision can generate strictly richer distributions. More specifically, a DLM with remasking or revision can generate the distribution with constant steps (Theorem 4.3), while it is impossible for a DLM with neither of them (Theorems 4.1 and 4.2). Our results show clear evidence that remasking and revision, which arise from the special structure of DLMs, can potentially greatly enhance the capability of the model given constrained inference time. This suggests exploring more remasking strategies and more flexible forward-processes in the future.

## 1.1 RELATED WORK

**Diffusion Language Model.** Diffusion models were originally built for generative modeling in the continuous domain (Ho et al., 2020; Sohl-Dickstein et al., 2015), and were later extended to the discrete domain. The seminal work of D3PM (Austin et al., 2021) formalized categorical forward kernels and popularized absorbing masks, and is followed by a series of works on language modeling (He et al., 2023; Lou et al., 2024; Sahoo et al., 2024; Shi et al., 2024). Furthermore, recent works (Song et al., 2025; Nie et al., 2025; Yang et al., 2025; Ye et al., 2025; Xie et al., 2025; Khanna et al., 2025) scale DLMs up, and achieve competitive performance to AR models and faster generation.

**Decoding order, Remasking and Revision.** DLM can perform inference in any order, but it is observed that decoding order can greatly affect the performance of DLMs (Kim et al., 2025; Yang et al., 2025; Xie et al., 2025). Remasking Nie et al. (2025); Yang et al. (2025) is another inference-time strategy that introduces richer decoding paths to further enhance the performance. More recently, DLMs with revision (Song et al., 2025) are introduced and exhibit remarkable capability. However, the fundamental differences of these strategies are not clear yet.

## 2 PRELIMINARY

In this section, we specify the notations and conventions used in this paper, and describe the theoretical framework to analyze the problems.

**Notations.** Let $\mathcal{V}$ be the vocabulary set. Throughout the paper, we will always consider the binary vocabulary set $\mathcal{V} = \{a, b\}$ by default, but the diffusion language model works for any other discrete vocabulary set. We also consider a *mask token* M. We consider sequences of length $L$ on the $\mathcal{V} \cup \{M\}$. For a sequence $x_t \in (\mathcal{V} \cup \{M\})^L$, $t \in [0, 1]$ denotes the noise level, i.e., the fraction of the masked tokens in the sequence. For simplicity, we assume $t \in \{0, 1/L, 2/L, \ldots, (L-1)/L, 1\}$, and therefore $t \cdot L$ is an integer. Let $[n] = \{1, \ldots, n\}$ for a positive integer $n$. A sequence with no mask is called noiseless. We use $x_t^i$ where $i \in [L]$ to denote the $i$-th entry of $x_t$, and $x_t^{i:j}$ to be the subsequence of $x_t$ starting from the $i$-th entry to the $j$-th entry, including both ends. Since subscripts are used as noise levels, we use superscripts for indices of sequences by default. We denote the set of distributions with support on set $S$ by $\Delta(S)$.

## 2.1 DIFFUSION LANGUAGE MODEL

Central to a diffusion language model (DLM) is a predictor $p(\cdot|x_t) \in \Delta(\mathcal{V}^L)$ that predicts a noiseless sequence from a noisy sequence $x_t \in (\mathcal{V} \cup \{M\})^L$. Let $p^i(\cdot|x_t)$ be the distribution on

the token at position $i$. The predictor satisfies that its prediction on tokens at different positions are conditionally independent given $x_t$, and it only changes mask tokens. Mathematically, for all $x_0 \in \mathcal{V}^L, x_t \in (\mathcal{V} \cup \{\mathrm{M}\})^L$, we have

$$p(x_0|x_t) = \prod_{i=1}^{L} p^i\left(x_0^i|x_t\right), \text{ and } p^i\left(x_t^i|x^t\right) = 1 \text{ if } x_t^i \in \mathcal{V}.$$

These constraints arise from the definition of the forward process, where tokens at different positions are independently masked to M.

At inference time, we specify a noise level $s$ that we want the output sequence to be and unmask a set of positions to match this noise level. Following the convention of Kim et al. (2025), we denote the set of positions to be unmasked by $\mathcal{S} = \mathcal{F}(x_t) \subseteq \left\{i|x_t^i = \mathrm{M}\right\}$ that depend on $x_t$. A simple unmasking policy $\mathcal{F}(x_t)$ only requires the knowledge of where the masked tokens are and chooses a uniformly random set $\mathcal{S} \subseteq \left\{i|x_t^i = \mathrm{M}\right\}$ to unmask, such that $|\mathcal{S}| = L(t-s)$. For much of the paper we constrain $\mathcal{F}$ to be a deterministic function.

We also consider the general case of DLMs trained to generate an output $o \in \mathcal{V}^m$ conditioned on an input $q \in \mathcal{V}^n$. At inference time, we start from $x_t = q\mathrm{M}^{L-n}$, the sequence of input padded with masks, and adopt an iterative generation process. In each iteration, we first calculate the positions to decode as $\mathcal{S} = \mathcal{F}(x_t)$, and then decode the corresponding positions using $p^i(\cdot|x)$ for $i \in \mathcal{S}$. The iteration ends at a predetermined number of steps $D$ at a noiseless sequence $x_0$. We finally extract the output as $o = x^{L-m+1:L}$. The generation process is summarized in Algorithm 1.

Chain of Thought (CoT) is the technique in language models that allows the output $o$ to a prompt $q$ to appear after generating some intermediate tokens. In our framework, this means that $L > m+n$, and the form of $x_0$ is $q|\mathrm{CoT}|o$. In this paper we always assume that $L$ matches the length of $q|\mathrm{CoT}|o$ for simplicity, while in practice $L$ is predetermined and is usually larger than the length of $q|\mathrm{CoT}|o$. We also allow for the case when $q$ is an empty string, which corresponds to generating an unconditional distribution.

**Remasking.** Remasking is an inference strategy that is used to improve output quality. It allows unmasked tokens to be remasked during the generation process. We model this via function $\mathcal{G}(\cdot)$, which at the end of each iteration, it produces a set $\mathcal{T} = \mathcal{G}(x_t) \subseteq [L]$ that represents the positions to be remasked and set to M. By default, we consider $\mathcal{G}$ that is a random function. The remasking process is also incorporated in Algorithm 1.

**Revision.** Recent works such as Song et al. (2025) have proposed forward processes that allows a token to change into another token before becoming fully masked. We refer to this as DLMs with *revision*. When DLM with revision are considered, we relax the constraint that an unmasked token must remain fixed. However, we still impose the constraint that token distributions across different positions are independent. Formally, for all $x_0 \in \mathcal{V}^L, x_t \in (\mathcal{V} \cup \{\mathrm{M}\})^L$, we have

$$p(x_0|x_t) = \prod_{i=1}^{L} p^i\left(x_0^i|x_t\right), \text{ but it is possible that } p^i\left(x_t^i|x^t\right) < 1 \text{ for } x_t^i \in \mathcal{V}.$$

Unless otherwise specified, throughout this paper DLM refers to the standard model without remasking or revision.

## 2.2 BOOLEAN CIRCUITS

A Boolean circuit is a model to characterize the complexity of carrying out certain parallel computations. A circuit $C$ is defined as a layered directed acyclic graph (DAG). Formally, a DAG $C = (V, E)$ consists of a set of vertices $V$ and a set of directed edges $E \subseteq V \times V$ with no directed cycles. The number of vertices $|\mathcal{V}|$ is called the size of the circuit. Furthermore, there is a function $l : V \to \mathbb{Z}^+$, which assigns every vertex to a layer satisfying $l(v_2) = l(v_1) + 1$ for all $(v_1, v_2) \in E$. That is, no edge skips a layer. Each vertex represents an *input*, a *logic gate* or an *output*. The input and output vertices have zero in-degree and out-degree, respectively, and they take Boolean values from $\{0, 1\}$. Logic gates perform a range of operations on their inputs, which we will specify in the following

---

**Algorithm 1** DLM Inference

---

**Input:** prompt $q \in \mathcal{V}^n$, $p(\cdot|x_t)$, length $L$, rounds $D$, masking $\mathcal{F}$, optional remasking $\mathcal{G}$.
**Output:** Output $o \in \mathcal{V}^m$

   $x \leftarrow q\mathrm{M}^{L-n}$
   **for** $j = 1$ to $D - 1$ **do**
      $\mathcal{S} \leftarrow \mathcal{F}(x)$.
      $x^i \sim p^i(\cdot|x)$ for each $i \in \mathcal{S}$.
      $\mathcal{T} \leftarrow \mathcal{G}(x)$.                          // If remasking is enabled
      $x^i \leftarrow \mathrm{M}$ for each $i \in \mathcal{T}$.               // If remasking is enabled
   **end for**
   $x^i \sim p^i(\cdot|x)$ for each remaining masked position.
   **return** $x^{L-m+1:L}$

---

paragraphs. As a result, we can calculate the output from the input using logic gates following the graph.

The input to the a circuit is $(\chi, R) \in \{0, 1\}^n \times \{0, 1\}^r$ and consists of two parts: $\chi \in \{0, 1\}^n$ represents the actual input, and $R \in \{0, 1\}^r$ are the random bits, sampled from $U^r = \text{Uniform}(\{0, 1\}^r)$, the uniform distribution over $\{0, 1\}^r$. We denote the output of the circuit as $C(\chi, R)$. The *depth $d$* and *width $w$* of the circuit are respectively defined as

$$d = \max_{v \in \mathcal{V}} l(v), w = \max_{\ell \in [d]} |\{v | v \in \mathcal{V}, l(v) = \ell\}|.$$

That is, depth is the total number of layers, and width is the largest number of vertices in a layer. Depth represents the minimum number of sequential steps to carry out the computation represented by $C$, and width represents the minimum number of memory bits needed to carry out the computation. Finally, we say that $C$ simulates a random function $f : \{0, 1\}^n \to \{0, 1\}$ if

$$\Pr[f(\chi) = \psi] = \Pr_{R \sim U^r}[C(\chi, R) = \psi] \text{ for all } \chi \in \{0, 1\}^n, \psi \in \{0, 1\}^m.$$

In this paper, we use Latin letters $x, y$ to represent token sequences, and Greek letters $\chi, \psi$ to represent boolean sequences. We use Boolean circuit to model the computational complexity to generate distributions. All distributions in this paper are assumed to be generated by some circuit. Furthermore, we also model the diffusion language model, including the predictor, $\mathcal{F}$ and $\mathcal{G}$, as circuits. Since the circuit model is a standard model of modern computers, it is natural to use circuits to model the distributions that can be sampled by computers, as well as the computation in DLMs. Besides, it has been standard to study the expressive power of neural networks using circuit models (Li et al., 2024; Merrill & Sabharwal, 2023).

**Complexity Classes.** A *problem* is a random map $\mathcal{L} : \{0, 1\}^* \to \{0, 1\}^m$. A set of circuits $\{C^n\}_{n \in \mathbb{N}}$ simulates problem $\mathcal{L}$ if

$$\Pr[\mathcal{L}(\chi) = \psi] = \Pr_{R \sim U^{r|\chi|}}\left[C^{|\chi|}(\chi, R) = \psi\right] \text{ for all } \chi \in \{0, 1\}^*, \psi \in \{0, 1\}^m.$$

Here $r^n$ represents the number of random bits in $C^n$.

A basic set of gates for circuits is $\{\mathsf{AND}, \mathsf{OR}, \mathsf{NOT}, \mathsf{ID}\}$. Gates $\mathsf{AND}, \mathsf{OR}$ has 2 in-degrees, while $\mathsf{NOT}, \mathsf{ID}$ has 1 in-degree. All of them have 1 out-degree. Gates $\mathsf{AND}, \mathsf{OR}, \mathsf{NOT}$ realize the logic computations suggested by their names, and $\mathsf{ID}$ outputs the input without change. A problem $\mathcal{L}$ is in $\mathsf{NC}^k$ where $k \in \mathbb{N}$, if there exists a set of circuit $\{C_n\}_{n \in \mathbb{N}}$, such that it computes $\mathcal{L}$, and any $C_n$ has $O(\text{poly}(n))$ gates and $O\left((\log n)^k\right)$ depth. Now we generalize the complexity classes to more powerful classes. If we allow $\mathsf{AND}, \mathsf{OR}$ to have unbounded in-degrees rather than 2 in-degrees, the corresponding complexity class is called $\mathsf{AC}^k$. If we further allow $\mathsf{MAJORITY}$ gates with unbounded in-degrees, which outputs 1 if and only if there are more 1's than 0's in its input, the corresponding complexity class is called $\mathsf{TC}^k$. $\mathsf{MAJORITY}$ could be thought of as allowing the circuit to count, as assessing which digit is majority essential requires the circuit to count their numbers of appearance. It is known that for all $i \in \mathbb{N}$, $\mathsf{NC}^i \subseteq \mathsf{AC}^i \subseteq \mathsf{TC}^i \subseteq \mathsf{NC}^{i+1}$ (Vollmer, 1999).

**Circuit Model for DLM.** Recall that we restrict the vocabulary[1] $\mathcal{V} = \{a, b\}$ to be binary. In order to encode the vocabulary as well as the masked token M, we will use a fixed 2-bit encoding

$$a \to 00, \quad b \to 01, \quad M \to 10,$$

with the unused encoding 11 being invalid and never produced. For a sequence $x \in (V \cup \{M\})^L$ let its encoding be $\chi \in \{0, 1\}^{2L}$, where the pair $(\chi_{2i-1}, \chi_{2i})$ encodes $x^i$. When convenient we refer to the numerical or token value of $a$ and $b$ as 0 and 1, rather than their encoded values 00 and 01.

A predictor $p(\cdot|x_t)$ of a DLM is a set of $L$ circuits whose inputs are from the encoding space. For each position $i \in [L]$ we have a circuit $C^i : \{0, 1\}^{2L} \times \{0, 1\}^{r_i} \to \{00, 01, 10\}$, which takes as input the shared encoded sequence $\chi_t$ and uses $r^i$ random bits $R^i \sim U^{r^i}$ to simulate $p_i(\cdot|x_t)$. Analogous to the definition of DLMs before, the random bits for different $i$ are independent so that $p(x_0|x_t) = \prod_{i=1}^{L} p^i\left(x_0^i \middle| x_t\right)$. Furthermore, each circuit $C_i$ can only unmask a masked token, i.e., if $\chi_t^{2i-1:2i} \neq 10$ then the output pair at $i$ equals the input pair.

Masking and remasking are done analogously. In particular, function $\mathcal{F}$ is represented by a deterministic boolean circuit $f : \{0, 1\}^{2L} \to \{0, 1\}^L$, where $f(\chi_t)^i$ represents whether the $i$-th position should be unmasked. Similarly, function $\mathcal{G}$ is represented by a possibly randomized circuit function $g : \{0, 1\}^{2L} \times \{0, 1\}^r \to \{0, 1\}^L$, where $r \sim U^r$ and $g(\chi_t, r)^i$ represents whether the $i$-th position should be unmasked.

## 3 DLMs ARE EFFICIENT SAMPLERS

In this section, we prove that DLMs can simulate circuits with the minimum possible parallel computation steps and memory needed. As mentioned in Section 2.2, the depth and width of a circuit provide a natural formalization of the minimum sequential steps and memory needed to sample from a distribution. In this section, we denote a DLM as $M$ and use $M(q)$ to represent the output from the DLM given input $q$ generated using the DLM following Algorithm 1.

### 3.1 DLM WITH CHAIN OF THOUGHT

The following theorem formalizes the statement that DLMs can generate any distribution with the minimum possible sequential steps.

**Theorem 3.1.** *Let $C$ be a circuit with depth $d$, $m$ output bits, $n$ input bits, random input bits $R \sim U^r$ and $N$ vertices. There is a DLM $M$ with CoT and length $L = N$, such that for every input $q$, $M(q)$ generates the same distribution as $C(q, R)$. Furthermore, the number of decoding steps of $M$ is $d$, and the corresponding predictor $p$ and unmask position policy $\mathcal{F}$ are both represented by constant-depth circuits.*

Before presenting the proof, we first define a function that is very useful for our construction.

**Definition 1.** *Boolean function* $\mathsf{ShiftR} : \{0, 1\}^n \to \{0, 1\}^n$ *is defined as follows. For any* $\chi \in \{0, 1\}^n$,

$$\mathsf{ShiftR}(\chi)^i = \begin{cases} \chi^{i-1} & \text{if } 1 < i \leq n \\ 1 & \text{if } i = 1 \end{cases}$$

Simulating this function is in $\mathsf{NC}^0$. In other words, this circuit can be realized in $O(1)$ with $O(\text{poly}(n))$ $\mathsf{AND}, \mathsf{OR}$ with 2 in-degree and $\mathsf{NOT}$ gates. The 1 can be realized by $1 = \eta \lor \neg\eta$ for any boolean variable $\eta$, and the others come directly from the input.

*Proof of Theorem 3.1.* Let us start by introducing some notations.

**Notations.** Without loss of generality, we can assume that all input vertices are in the first layer, all output vertices are in the last layer, and there is no random bit in the first layer. We define the following variables:

---

[1]Larger vocabularies are handled by standard reduction that encodes tokens by binary codewords and expand the sequence length accordingly.

- $v^1, \ldots, v^N$ be the list of vertices in the circuit; they are sorted in a way that the layer number increases in the sequence,

- $u^1, \ldots, u^N$ be the corresponding variables that are output from the vertices;

- $l^1, \ldots, l^N$ be the layer number of each vertex;

- $s^1, \ldots, s^d$ be the index of the first vertex in every layer (and $s^0 = 0, s^{d+1} = L + 1$ for simplicity);

- $w^1, \cdots, w^d$ be the width of each layer in $C$, i.e. $w^i = s^{i+1} - s^i$ for all $i = 1, \ldots, d$.

In this proof, *we use the subscript of $x$ to index the iteration that is being carried out instead of the noise level for convenience.* For example, the initial sequence for the model is $x_1 = q\mathrm{M}^{L-n}$, and becomes $x_2$ after the first iteration. **The generation process.** Now let us describe the generation process that we wish to realize. Our construction carries out the computation of circuits in layer $i$ during iteration $i$, appends the output to the end of existing outputs, and carries out the computation for layer $i + 1$ based on it. Mathematically, we realize the following intermediate sequences:

$$x_i^j = \begin{cases} u^j & \text{if } j < s^{i+1} \\ \mathrm{M} & \text{if } j \geq s^{i+1} \end{cases}.$$

As a result, the generation process stops at sequence $x_d = u^1 \cdots u^N$ and $x_d^{L-m+1:L}$ equals $o$. Below we will describe how to construct the DLM $M$ (specifically, the functions $\mathcal{F}$ and the predictor $p$) that realizes this generation process.

**Construction of $\mathcal{F}$.** To implement the process above, function $\mathcal{F}$ needs to identify the current iteration index $i$ from $x_i$, and produce $\mathcal{S} = \left\{s^{i+1}, \ldots, s^{i+2} - 1\right\}$ accordingly. Since $\chi_i^{2j-1}$ denotes whether $x_i^j$ is a mask according to the token encoding, it follows that the sequence $\left\{\chi_i^{2s^j-1}\right\}_{j=1,\ldots,d}$ denotes whether each layer in the circuit is decoded in $x_i$. Let $\chi_i^{2s-1} \in \{0,1\}^d$ be the concatenation of this sequence, we have that $(\chi_i^{2s-1})^j$ and $\mathsf{ShiftR}\left(\chi_i^{2s-1}\right)^j$ denote whether layer $j$ in the circuit is decoded in $x_i$ and $x_{i+1}$, respectively. Hence $\left(\mathsf{ShiftR}\left(\chi_i^{2s-1}\right) \wedge \neg\chi_i^{2s-1}\right)^j$ denotes whether layer $j$ should be decoded at iteration $i$ from $x_i$. In other words, it equals $1$ only when $j = i + 1$. Putting everything together, we set

$$f(\chi_i)^j = \left(\mathsf{ShiftR}\left(\chi_i^{2s-1}\right) \wedge \neg\chi_i^{2s-1}\right)^{l^j},$$

which is realized by a constant-depth circuit using $\mathsf{OR}$ with 2 in-degree and $\mathsf{NOT}$.

**Construction of $p(\cdot|x_t)$.** To implement the process above, our $p(\cdot|x_t)$ only needs to replicate the computation from the $i$-th layer to the $(i+1)$-th layer. Notice that the construction of $p(\cdot|x_t)$ is identical for all iterations. *The layer of the circuit to simulate is controlled by $\mathcal{F}$, which extracts the information of $i$ from the mask patterns in $x_t$ as described in the last paragraph.* Denote the circuit that computes the outputs from the $i$-th layer to the $(i+1)$-th layer by $C_i : \mathcal{V}^{w^i} \times \{0,1\}^{r^{i+1}} \to \{0,1\}^{w^{i+1}}$. [2] Here $r^{i+1}$ represents the number of random bits in layer $i + 1$. Then the output for $p(\cdot|x)$ is: for $s^{i+1} \leq j \leq s^{i+2} - 1$

$$\chi_{i+1}^{2j-1} = 0, \chi_{i+1}^{2j} = C_i\left(x_i^{s^i:s^{i+1}-1}, R_i\right)^j, R_i \sim U^{r^{i+1}}.$$

Output bits for other $j$ are not specified because other positions are not unmasked and are hence not relevant.

Finally, we need to confirm that the outputs of $p(\cdot|x_t)$ at different positions are independent. Note that in $C_i$, the random bits $R_i$ is only used for generating the random bits in the $(i+1)$-th layer. As a result, every bit of $R$ becomes a random bit in the $(i+1)$-th layer, and is not involved in the calculation of any other output bit. Hence, the constructed $p(\cdot|x_t)$ indeed realizes an entry-wise independent output distribution.

□

---

[2]To simplify notation, here we assume that the circuit takes in tokens from $\mathcal{V}$, which is mapped to the binary representation before being passed to the actual circuit

In the construction, we have to record intermediate calculation results (sequence $u$) as each application of $p$ can only simulate a constant-depth circuit. Meanwhile, we cannot erase previous intermediate results without remasking or revision, despite only needing them once. As a result, though the time complexity is amenable, the construction incurs a large memory footprint.

This result demonstrates that DLMs can in principle generate distributions faster than their autoregressive counterparts and hold great potential for fast parallel generation. However, in practice, being able to sample in $O(d)$ rounds instead of $O(N)$ does not directly imply that we could reduce computation or wall-clock latency compared to autoregressive models in real-world practice. As a concrete example, we compare the computation cost and wall-clock latency in one decoding step between an autoregressive Transformer with KV caching and a DLM implemented by a Transformer of the same size. In every attention layer, an autoregressive model only needs to compute the attention output for one position, whereas a DLM needs to compute the attention output for all positions. Hence, a DLM needs $O(L)$ times more FLOPs than an autoregressive model. The computation of the $L$ tokens is parallelizable for the DLM, so *if the hardware supports enough degree of parallelism*, they have the same wall-clock latency. However, if the hardware does not support enough parallel computing, the wall-clock latency for the DLM can be slower.

## 3.2 DLM with Chain of Thought and Remasking or Revision

With remasking or revision, we have the power to erase intermediate results that are no longer needed. In this part, we show that with remasking or revision, we can sample from any distribution not only with the minimum possible sequential steps, but also with the minimum amount of memory. The proofs of the following two theorems are deferred to the Appendix as the proof strategy is similar to that of Theorem 3.1

**Theorem 3.2.** *Let $C$ be a circuit with depth $d$, width $w$, $m$ output bits, $n$ input bits, and random input bits $R \sim U^r$. There exists a DLM $M$ with CoT and remasking with length $L = 2w + 2\lceil \log(d+1) \rceil$, such that the distribution of $M(q)$ is the same as $C(q, R)$. Furthermore, the number of decoding steps is $d + 1$, and $p, \mathcal{F}, \mathcal{G}$ are all represented by circuits with $O(\log d)$ depth.*

The $\lceil \log(d+1) \rceil$ term in the length $L$ comes from the fact that we can no longer infer the iteration step from the mask pattern, and need extra spaces to record the current iteration step. The proof is simpler for DLMs with revision, as we can directly rewrite $x_i$ by $x_{i+1}$, instead of using two chunks of memory in an alternating way. The theorem statement is as follows:

**Theorem 3.3.** *Let $C$ be a circuit with depth $d$, width $w$, $m$ output bits, $n$ input bits, and random input bits $R \sim U^r$. There exists a DLM $M$ with CoT and revision with length $L = w + \lceil \log(d+1) \rceil$, such that the distribution of $M(q)$ is the same as $C(q, R)$. Furthermore, the number of decoding steps is $d + 1$, and $p, \mathcal{F}$ are both represented by circuits with $O(\log d)$ depth.*

## 4 DLMs are More Expressive with Remasking or Revision

In the last section, we show how DLMs with appropriate sizes can efficiently simulate the generation process of Boolean circuits, and how enabling remasking or revision can significantly reduce the sequence length $L$ needed. It is natural to ask: for a DLM with fixed $L$, does remasking or revision enable us to generate new distributions with constant generation steps? This section gives an affirmative answer to this question by showing a separation on generating a specific distribution introduced in the next paragraph.

For the purpose of showing separation, we only need to consider the simpler case when the DLM takes no input ($q = \varnothing$), and is sampling from an unconditional distribution. Let $\oplus$ be addition modulo 2. That is, $z_1 \oplus z_2 = b$ for $b \in \{0, 1\}$ iff $z_1 + z_2 = b \pmod 2$. It is well-known that $\oplus$ can be represented by AND and OR as

$$\chi \oplus \psi = (\chi \vee \psi) \wedge \neg(\chi \wedge \psi) \quad \text{for any } \chi, \psi \in \{0, 1\}$$

We define the parity function as follows.

**Definition 2.** *Boolean function* $\mathsf{PARITY} : \{0,1\}^n \to \{0,1\}$ *is defined as follows. For any* $\chi \in \{0,1\}^n$,

$$\mathsf{PARITY}(\chi) = \bigoplus_{i=1}^{n} \chi_i.$$

It is well known that $\mathsf{PARITY}$ is not in $\mathsf{AC}^0$ (Furst et al., 1984; Håstad, 1986). We show the separation result using the following distribution.

**Definition 3.** *Distribution* $\mathcal{D}_n^{\oplus}$ *is defined as the uniform distribution over all $n$-bit strings with even parity (i.e.,* $\{(z_1, \ldots, z_n) | \mathsf{PARITY}(z) = 0\}$*).*

In this section, we assume that DLMs consist of $\mathsf{AC}^0$ circuits. Note that constant-precision constant-depth Transformers with poly-sized embedding size can be simulated by $\mathsf{AC}^0$ circuits Li et al. (2024), and most current DLM implementations are based on Transformers Nie et al. (2025); Yang et al. (2025). Besides, Transformers with hard attention mechanism, where each attention block can only attend to the position with the largest score, can be simulated by $\mathsf{AC}^0$ circuits as well Hao et al. (2022). Hence, we believe that our results reflect the true expressiveness of current architectures. In particular, for the task of sampling from $\mathcal{D}_n^{\oplus}$, the sampling algorithm described above is very simple and can be easily simulated by Transformers.

### 4.1 UPPER BOUND

In this part, we show that DLMs with revision can sample strings from $\mathcal{D}_n^{\oplus}$ in two steps.

**Theorem 4.1.** *DLMs with revision and $L = n$ can generate $\mathcal{D}_n^{\oplus}$ in two steps, such that the predictor $p$ and function $\mathcal{F}$ are both represented by $\mathsf{NC}^0$ circuits.*

*Proof.* Like the proof of Theorem 3.1, we first describe the generation process that we wish to realize, and then show the circuit construction. We also keep the notation that $x_i$ represents the sequence after the $i$-th generation step.

Our goal is to generate $(z^1, \ldots, z^n) \sim \mathcal{D}_n^{\oplus}$. Let $(y^1, \ldots, y^n) \in \{0,1\}^n$ be defined as $y^i = \bigoplus_{j=1}^{i} z^j$. It is easy to verify that $(y^1, \ldots, y^{n-1}) \sim U^{n-1}$, and $y^n = 0$. Furthermore, $z^1 = y^1$ and $z^i = y^i \oplus y^{i-1}$ for $i = 2, \ldots, n$ by definition, so we can easily recover $z$ from $y$. Hence we may generate $\mathcal{D}^{\oplus}$ with the following process: $x_0 = \mathsf{M}^n$, $x_1 = y$, $x_2 = z$. The procedure above can be implemented by DLMs with revision easily: In the first step, we generate $(y^1, \ldots, y^{n-1}) \sim U^{n-1}$, and $y^n = 0$. In the second step, we calculate $z^1 = y^1$ and $z^i = y^i \oplus y^{i-1}$ for $i = 2, \ldots, n$. Below, we describe the circuit construction in detail.

Function $\mathcal{F}$ always outputs an all-one vector of length $n$. Formally $f(\chi)^j = 1$ for all $\chi \in \{0,1\}^{2n}$.

Now let us describe the circuit for $p(\cdot | x_t)$. We need to first identify whether we are in $i = 0$ or $i = 1$ by looking at whether the first token is masked ($\chi^1$), and then carry out the corresponding computations. Hence, for every $i \in \{0, 1\}$,

$$\chi_{i+1}^{2j-1} = 0, \chi_{i+1}^{2j} = \begin{cases} \left(\chi_i^1 \wedge R^1\right) \vee \left(\neg\chi_i^1 \wedge \chi_i^2\right) & \text{if } j = 1 \\ \left(\chi_i^1 \wedge R^j\right) \vee \left(\neg\chi_i^1 \wedge \left(\chi_i^{2j} \oplus \chi_i^{2j-2}\right)\right) & \text{if } 1 < j < n \\ \left(\chi_i^1 \wedge 0\right) \vee \left(\neg\chi_i^1 \wedge \left(\chi_i^{2j} \oplus \chi_i^{2j-2}\right)\right) & \text{if } j = n \end{cases}, R \sim U^{n-1}.$$

The above shows $p(\cdot | x_t)$ can be implemented easily by a constant-depth circuit (the constant-depth circuit above maps $\chi_i$ to $\chi_{i+1}$ for $i \in \{0, 1\}$). Also, note that every position of $\chi$ depends on at most one random input bit, so the outputs for different positions are independent conditioned on $\chi_i$. This completes the proof. $\square$

A similar result holds for DLMs with remasking, as stated below.

**Theorem 4.2.** *DLMs with remasking and $L = n$ can sample $\mathcal{D}_n^{\oplus}$ in $O(1)$ steps, such that the corresponding circuits are in $\mathsf{AC}^0$. Here, we also allow the unmask position predictor $\mathcal{F}$ to depend on the step number in the generation process.*

*Proof.* In the following, we describe the sampling algorithm while also explaining how to implement it using an $\mathsf{AC}^0$ predictor $p$ and an unmask position predictor $\mathcal{F}$. Since here our predictor $p$ and $\mathcal{F}$ can also depend on the step number in the generation process, at each step we can simply use a new $p$ and $\mathcal{F}$.

Suppose for simplicity that $n$ is even. We will proceed in two stages, each consisting of $O(1)$ steps. First, we partition the $n$-bit input into $n/2$ 2-bit blocks consecutively. In the first stage, at each even position $2i$, we will sample a bit $x^i$ such that the parity of the positions at $2i-1$ and $2i$ should be $x^i$. Note that all these $x^i$ should follow the distribution $\mathcal{D}^{\oplus}_{n/2}$. Next in the second stage, we will sample each 2-bit block (i.e., $(2i-1, 2i)$ for $i = 1, \ldots, n/2$) uniformly at random such that its parity is $x^i$. This gives us the desired distribution $\mathcal{D}^{\oplus}_n$.

**Stage 1. Sampling the parity of $n/2$ pairs.** For this stage, we will proceed in two steps. In the first step, at all odd positions, we generate $y^1, y^2, \ldots, y^{n/2}$ such that $(y^1, \ldots, y^{n/2-1})$ is uniformly random and $y^{n/2} = 0$. Then, in the second step, at all even positions, we compute $x^i = y^i \oplus y^{i-1}$ for $i \in \{2, \ldots, n/2\}$, and $x^1 = y^1$. After these two steps, we have generated $(y^1, x^1, y^2, x^2, \ldots, y^{n/2}, x^{n/2})$. The first two steps can be implemented using $\mathsf{AC}^0$ predictor $p$ and $\mathcal{F}$ easily.

Now, following the proof of Theorem 4.1, $(x^1, \ldots, x^{n/2})$ follows the distribution $\mathcal{D}^{\oplus}_{n/2}$, meaning that they are uniformly random conditioning on having even parity.

**Stage 2. Sampling all the $n/2$ pairs given their parities.** Next, we remask all odd positions (all the $y^i$'s). Recall that we interpret $x^i$ as the parity of the $i$-th 2-bit block. The goal from now on is to sample the $i$-th 2-bit block $(2i-1, 2i)$ uniformly at random such that its parity is $x^i$, which gives us the desired distribution $\mathcal{D}^{\oplus}_n$.

Concretely, for each block of the form M0, we will replace it with a new string from $\{00, 11\}$ with equal probability. For each block of the form M1, we will replace it with a new string from $\{01, 10\}$ with equal probability. We will do it in two sub-stages, each with several steps.

**Stage 2.a. Dealing with blocks of the form M0.** In the first sub-stage, for all blocks of the form M0, we sample the first bit randomly. Then, for every block of the form 10, we first remask the 0 to get 1M, and then change the last M to 1 to get 11. In this way, each M0 block is replaced by 00 or 11 with equal probability as desired.

**Stage 2.b. Dealing with blocks of the form M1.** In the second sub-stage, for all blocks of the form M1, we sample the first bit randomly. Then, for every block of the form 11, we first remask the second 1 to get 1M, and then change the last M to 0 to get 10. In this way, each M1 block is replaced by 01 or 10 with equal probability as desired.

It is not hard to see that the two sub-stages can be implemented using an $\mathsf{AC}^0$ predictor $p$ and $\mathcal{F}$ easily if the predictor and unmask position predictor can also depend on the step number in the generation process. This completes the proof. $\qquad\square$

## 4.2 LOWER BOUND

In this part, we show that DLMs without remasking or revision cannot generate $\mathcal{D}^{\oplus}_n$ in $O(1)$ steps even if we allow the predictor $p$ and $\mathcal{F}$ to be $\mathsf{AC}^0$ circuits. In this way, we establish a strong separation between DLMs with and without revision/remasking.

This result should come across as intuitive, given that PARITY is not in $\mathsf{AC}^0$ as mentioned before. During DLM generation, one can only modify mask tokens. When there is only one mask token left, the DLM has to calculate the parity of all other bits, which is not possible. However, formalizing this intuition precisely is challenging because DLM can decode multiple tokens simultaneously in each iteration. As a result, we need an average-case lower bound for parity against $\mathsf{AC}^0$ as introduced below.

**Theorem 4.3** (Håstad (2014)). *Let $d$ be a constant, and $C \colon \{0,1\}^n \to \{0,1\}$ be an $\mathsf{AC}^0$ circuit of depth $d$ and size $S$. There exists a constant $c_d$ that only depends on $d$ such that*

$$\Pr_{\chi \sim U^n}[C(\chi) = \mathsf{PARITY}(\chi)] < 1/2 + 2^{-c_d \cdot n/(\log S)^{d-1}}.$$

In plain text, this theorem says that any $\mathsf{AC}^0$ circuit cannot calculate PARITY better than random guessing by too much. Furthermore, this theorem can be directly extended to probabilistic $\mathsf{AC}^0$ circuits as follows.

**Corollary 4.4.** *Let $d$ be a constant. Let $C\colon \{0,1\}^n \times \{0,1\}^m \to \{0,1\}$ be a probabilistic $\mathsf{AC}^0$ circuit of depth $d$ and size $S$. There exists a constant $c_d$ that only depends on $d$ such that*

$$\Pr_{\chi \sim U^n, R \sim U^m}[C(\chi, R) = \mathsf{PARITY}(\chi)] < 1/2 + 2^{-c_d \cdot n/(\log S)^{d-1}}.$$

Now we are ready to present the lower bound for DLM generations.

**Theorem 4.5.** *DLMs (without remasking or revision) with $L = n$ cannot sample $\mathcal{D}_n^\oplus$ in $O(1)$ steps when the predictor $p(\cdot|x)$ and $\mathcal{F}$ are both in $\mathsf{AC}^0$.*

*Proof.* Assume for the sake of contradiction that for any integer $n$, there exists a DLM that samples $\mathcal{D}_n^\oplus$ in $\ell$ steps such that the predictor $p$ and $\mathcal{F}$ are both in $\mathsf{AC}^0$.

We first set up some notations. Let $\ell = \Theta(1)$ be an arbitrary constant. We define the size of the DLM to be the maximum circuit size of the predictor $p$ (for all $i$) and $\mathcal{F}$. Since both the predictor $p$ and $\mathcal{F}$ are in $\mathsf{AC}^0$, the size of a DLM is upper bounded by $n^c$ for some constant $c > 1$, as long as $n$ is sufficiently large.

**Reduction to an Auxiliary Claim.** Under the assumption above, we can prove the following claim, which is instrumental for proving our theorem:

*For every $i \in \{0, 1, \ldots, \ell\}$, there exists an $O\left(n^c\right)$-size DLM with $L = n_i$ that samples $\mathcal{D}_{n_i}^\oplus$ in $\ell - i$ steps, where $n_i \geq n^{2^{-i}}$.*

Before proving the claim, we first show how this claim helps us finish the proof. This statement implies that an $n^c$-size DLM can sample $\mathcal{D}_{n^{2^{-\ell}}}^\oplus$ in $0$ step, a clear contradiction since one cannot sample any token in $0$ steps. The proof of the claim is deferred to the Appendix due to space limit.

$\square$

*Remark.* The proof of this theorem includes a procedure that converts a constant-round generation of an $\mathsf{AC}^0$ DLM to an $\mathsf{AC}^0$ circuit. One might ask the following question: Why cannot we use the same procedure to obtain an $\mathsf{AC}^0$ circuit from the construction in the proofs of Theorems 4.1 and 4.2 that samples $\mathcal{D}_n^\oplus$ in a single step? Does not this show that DLM without revision/remasking could sample $\mathcal{D}_n^\oplus$ in a single step? Indeed, one could obtain such $\mathsf{AC}^0$ circuit, but such $\mathsf{AC}^0$ circuit is not a valid DLM. A valid DLM predictor $p$ satisfies $p(x|x_t) = \prod_{i=1}^L p^i\left(x^i|x_t\right)$, so it is impossible to sample a distribution with correlation within a single step.

Our theoretical analysis advocates that designing a forward process that includes revision and collecting training data that contains revision/remasking is a very important design component for DLMs and holds greater potential than DLMs without revision or remasking.

## 5 CONCLUSION

In this paper, we establish a theoretical foundation for diffusion language models (DLMs) as parallel samplers, showing that with CoT they achieve the optimal number of sequential steps, in contrast to autoregressive models whose cost grows with circuit size. Furthermore, with remasking or revision, it can simultaneously achieve optimal space requirement. Additionally, we prove that remasking and revision not only reduce the memory requirement to scale with circuit width but also strictly expand expressivity, enabling constant-step sampling of distributions such as parity that standard DLMs cannot realize. These results position DLMs as the most efficient parallel samplers, and highlight remasking and revision as essential mechanisms for unlocking their full potential.

## 6 REPRODUCIBILITY STATEMENT

We state the assumptions in the main text and include proofs for our theoretical results in the Appendix.

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

## A  MISSING PROOFS FROM SECTION 3

**Theorem 3.2.** *Let $C$ be a circuit with depth $d$, width $w$, $m$ output bits, $n$ input bits, and random input bits $R \sim U^r$. There exists a DLM $M$ with CoT and remasking with length $L = 2w + 2\lceil \log(d+1) \rceil$, such that the distribution of $M(q)$ is the same as $C(q, R)$. Furthermore, the number of decoding steps is $d + 1$, and $p, \mathcal{F}, \mathcal{G}$ are all represented by circuits with $O(\log d)$ depth.*

**Some useful functions.** Again, we first introduce some functions that are useful for the construction, and then show that they can be realized in constant depth.

- Function $\text{bin} : \mathbb{N} \to \{a, b\}^{\lceil \log(d+1) \rceil}$ is a function that turns a natural number into its binary form.

- Boolean function $\text{ADD} : \mathcal{V}^d \to \{0, 1\}^n$ is defined as the function that outputs the binary number represented by the input plus one. When input is $b^d$, the output is $0^d$.

- Boolean function $\text{IDENTIFY}_i : \{a, b\}^d \to \{0, 1\}$ where integer $i < 2^d$ is defined as outputting 1 if and only if the input is $\text{bin}(i)$.

Note that ADD is in $\text{NC}^1$, following from a well-known result in circuit complexity (Brent & Kung, 1982). This means that the AND logic with $d$ inputs can be realized by AND with 2 in-degrees in $O(\log d)$ depth. The same holds for $\text{IDENTIFY}_i$, which follows from its alternative definition:

$$\text{IDENTIFY}_i(x) = \bigwedge_{j=1}^{d} \left( \chi^j \wedge \text{bin}(i)^j \right) \vee \left( \neg \chi^j \wedge \neg \text{bin}(i)^j \right).$$

Note that $O(\log d)$ is still constant with regard to $n$, so using these two circuits do not violate our restriction on constant-depth circuits.

*Proof of Theorem 3.2.* In this proof, we use the same notations as those of Theorem 3.1 as the proof strategy is similar. Again, we assume that all input vertices are in the first layer, all output vertices are in the last layer, and there is no random bit in the first layer. Furthermore, we assume that $d$ is even without loss of generality.

Before describing the generation process, we first partition $x_i$ into two blocks, each with length $w + d^*$. Here $d^* = \lceil \log(d + 1) \rceil$. With $d^*$ bits, we can record a layer index. Each block is used to record the output from a layer, together with the binary encoding of the layer index. These two blocks are used in an alternating way to store the output from the current layer and that of the next layer. Similar to the construction in Theorem 3.1, we simulate the calculation of one layer in each iteration.

**The generation process.** Now we formally describe the generation process that we wish to realize. As before, we use the subscript of $x$ to denote the iteration index. In the beginning, we have $x_0 = q \mathbf{M}^{L-n}$. For $0 < i \leq d$, we set

$$x_i = u^{s^i : s^{i+1} - 1} \mathbf{M}^{w - w^i} \text{bin}(i) \mathbf{M}^{w + d^*} \qquad \text{when } i \text{ is odd,}$$

$$x_i = \mathbf{M}^{w + d^*} \text{bin}(i) \mathbf{M}^{w - w^i} u^{s^i : s^{i+1} - 1} \qquad \text{when } i \text{ is even,}$$

In every iteration, we calculate the output of the next layer based on the output of the current layer and write it in the masked block. After that, we remask the block, recording the current layer output to reserve space for the next iteration output. In the end, the DLM outputs $o = x_d^{L - m + 1 : L}$.

**A useful construction of multiplexers.** Before delving into the construction detail, let us introduce a structure that appears many times: we need to construct a circuit that selects which circuit to use based on current layer index $i$, i.e. a multiplexer. Formally, we have circuits $D_1, \ldots, D_d$ with single outputs, and we want to construct a circuit $D$ such that outputs $D(x, \text{bin}(i)) = D_i(x)$. This can be realized with an $O(\log d)$ overhead in circuit depth and $O(\text{poly}(d))$ gates because we may construct

$$D(x, \text{bin}(i)) = \bigvee_{k=1}^{d} \text{IDENTIFY}_k(i) \wedge D_i(x). \tag{$\star$}$$

Here the OR logic with $d$ inputs can be realized by OR with 2 in-degrees in $O(\log d)$ depth and IDENTIFY has $O(\log d)$ depth. In this proof, $\mathrm{bin}(i)$ is either $x^{w+1:w+d^*}$ or $x^{w+d^*+1:w+2d^*}$, whichever is not masked. Formally, for $j = 1, \ldots, d^*$,

$$\mathrm{bin}(i)^j = \left(\chi^1 \wedge \chi^{2(w+d+j)}\right) \vee \left(\neg\chi^1 \wedge \chi^{2(w+j)}\right).$$

*As a result, in the rest of the proof, when we need a multiplexer, we only need to construct $D_i$ separately.*

**Construction of $\mathcal{F}, \mathcal{G}$ and $p(\cdot|x_t)$.** We first construct $\mathcal{F}$. When $i = 0$, we need the set of unmask positions $\mathcal{S}_0 = \{w + 1, \cdots, w + d^*\}$. For $0 < i < d$, we need the set of unmask positions

$$\mathcal{S}_i = \{w + d^* + 1, \ldots, w + 2d^*, L - w^{i+1} + 1, \ldots, L\} \qquad \text{when } i \text{ is odd,}$$
$$\mathcal{S}_i = \{1, \ldots, w^{i+1}, w + 1, \ldots, w + d^*\} \qquad \text{when } i \text{ is even.}$$

Formally, let $D_i(x) = \mathbb{1}\{x \in \mathcal{S}_i\}$, then we could use the multiplexer constructed above to realize the desired $\mathcal{F}$. Such multiplexer usage is similar for the construction of $\mathcal{G}$ and $p$.

Constructing $\mathcal{G}$ is similar to constructing $\mathcal{F}$. When $i = 0$, $\mathcal{T} = \varnothing$. For $0 < i < d$,

$$\mathcal{T} = \{1, \ldots, w + d^*\} \qquad \text{if } i \text{ is odd,}$$
$$\mathcal{T} = \{w + d^* + 1, \ldots, L\} \qquad \text{if } i \text{ is even.}$$

Finally, we construct $p$. Like the proof of Theorem 3.1, we only need to construct the output positions inside $\mathcal{S}$. Again, we use $C_i$ to denote the circuit from the $i$-th layer token outputs to the $(i + 1)$-th layer bit outputs. When $i = 0$, we have

$$x^{w+1:w+d} = \mathrm{bin}(1).$$

For $0 < i \leq d$ and $i$ is odd we have

$$\chi_{i+1}^{2j-1} = 0, \chi_{i+1}^{2j} = \begin{cases} C_i\left(x_i^{1:w^i}, R_i\right)^{j-L+w^{i+1}}, & \text{if } L - w^{i+1} + 1 \leq j \leq L \\ \mathrm{ADD}\left(x_i^{w+1:w+d^*}\right)^{j-w-d^*}, & \text{if } w + d^* + 1 \leq j \leq w + 2d^* \end{cases}, R_i \sim U^{w^{i+1}}.$$

This simulates the circuit computation from the $i$-th layer to the $(i + 1)$-th layer, and increments the iteration index. Similarly, for $0 < i \leq d$ and $i$ is even we have

$$\chi_{i+1}^{2j-1} = 0, \chi_{i+1}^{2j} = \begin{cases} C_i\left(x_i^{L-w^i+1:L}, R_i\right)^j, & \text{if } 1 \leq j \leq w^{i+1} \\ \mathrm{ADD}\left(x_i^{w+d+1:w+d+d^*}\right)^{j-w}, & \text{if } w + 1 \leq j \leq w + d^* \end{cases}, R_i \sim U^{w^{i+1}}.$$

$\square$

# B  MISSING PROOFS FROM SECTION 4

**Notation for the Claim Proof.** Let us now show how to prove the claim by induction. The case when $i = 0$ follows directly from our assumption. Now, assume that the claim holds for $i$, and we prove it for $i + 1$. Here we use similar notations as the proof of Theorem 3.1. Let $\mathcal{S}_1 = \mathcal{F}(\mathbf{M}^{n_i})$ be the set of positions to be unmasked in the first iteration, and $x_2^{\mathcal{S}_1} \in \{a, b\}^{|\mathcal{S}_1|}$ be the sequence of sampled tokens in the first iteration. Since the marginal distribution of $\mathcal{D}_{n_i}^{\oplus}$ on $\mathcal{S}_1$ is $U^{|\mathcal{S}_1|}$, and a DLM without revision or remasking cannot change unmasked tokens, we must have $x_2^{\mathcal{S}_1} \sim U^{|\mathcal{S}_1|}$. Furthermore, let $m = n_i - |\mathcal{S}_1|$ be the number of masked tokens in $x_2$. Since $U^{n_i} \neq \mathcal{D}_{n_i}^{\oplus}$, we must have $m > 0$.

**High-Level idea of the Claim Proof.** Before going into the details, here we introduce the high-level idea. The central argument is to show that $m \geq \sqrt{n_i}$ for sufficiently large $n_i$. To prove this, we will show that we can construct an $\mathrm{AC}^0$ circuit that approximates $\mathrm{PARITY}(\chi)$ for $\chi \in \{0, 1\}^{|\mathcal{S}_1|}$, and

the approximation becomes more accurate as $m$ is smaller. The circuit lower bound in Theorem 4.4 would then give us a lower bound on $m$ as an $\mathsf{AC}^0$ circuit cannot approximate PARITY very well. Besides, we can construct a DLM that samples $\mathcal{D}_m^\oplus$ in $\ell - i - 1$ steps using a DLM that samples $\mathcal{D}_{n_i}^\oplus$ in $\ell - i$ steps. Combined with $m \geq \sqrt{n_i}$ we finish the induction step.

**Key Construction in the Claim Proof.** The construction of a circuit $C$ that approximates PARITY $(\chi)$ for $\chi \in \{0,1\}^{|\mathcal{S}_1|}$ is as follows:

- Set the positions from set $\mathcal{S}_1$ in $x_2$ by $\chi$ using the correspondence $0 \mapsto a, 1 \mapsto b$. Set other positions of $x_2$ to be M.

- Simulate the DLM starting from $x_2$ for $\ell - i - 1$ steps and obtain output tokens on $[n_i] \setminus \mathcal{S}_1$.

- The DLM outputs sequence $x_{\ell-i}$. Circuit $C$ Output 0 if $x_{\ell-i}^{[n_i] \setminus \mathcal{S}_1}$ are all $a$, and output a random bit otherwise.

This circuit is in $\mathsf{AC}^0$ because every iteration we apply an $\mathsf{AC}^0$ circuit on $x$ and $\ell = O(1)$. Besides, outputting whether the outputs are all $a$ can be realized by an AND gate with unbounded fan-ins. Since parity of the output sequence PARITY$(x_{l-i}) = 0$ by definition of $\mathcal{D}^\oplus$, we know PARITY $\left( x_{\ell-i}^{\mathcal{S}_1} \right)$ = PARITY $\left( x_{\ell-i}^{[n_i] \setminus \mathcal{S}_1} \right)$. As a result, the constructed circuit calculates parity slightly better than a random guess on average by precisely predicting the parity in the case when $x_{\ell-i}^{[n_i] \setminus \mathcal{S}_1}$ is all $a$, as we must have PARITY $\left( x^{\mathcal{S}_1} \right) = 0$.

Now we calculate the probability that $C$ computes parity correctly over input $\chi \sim U^{|\mathcal{S}_1|}$. It is easy to verify that the distribution of $x^{[n_i] \setminus \mathcal{S}_1}$ conditioned on $x^{\mathcal{S}_1}$ with parity zero is $\mathcal{D}_{n_i - |\mathcal{S}_1|}^\oplus$. Hence the probability of $x_{\ell-i}^{[n_i] \setminus \mathcal{S}_1}$ being all $a$ is $2^{-m}$. Therefore the probability that $C$ outputs the parity correctly is

$$2^{-m} \cdot 1 + \left( 1 - 2^{-m} \right) \cdot \frac{1}{2} = \frac{1}{2} + 2^{-(m+1)}.$$

Note that $C$ has constant depth and size $n_i^c$, applying Theorem 4.4 we know that we must have

$$\frac{1}{2} + 2^{-(m+1)} \leq \frac{1}{2} + 2^{-\frac{c_d n_i}{(\log n_i^c)^{d-1}}} \Leftrightarrow m > \frac{c_d(n_i - m)}{c^{d-1} \left( \log(n_i - m) \right)^{d-1}} - 1.$$

For sufficiently large $n_i$, we have $m > \sqrt{n_i}$.[3] Finally, since the DLM samples perfectly from $\mathcal{D}_{n_i}^\oplus$, when setting $w_1$ to be all-zero, it samples $\mathcal{D}_m^\oplus$ over the positions in $[n_i] \setminus \mathcal{S}_1$. In this way, we obtain a DLM that samples from $\mathcal{D}_m^\oplus$ in $\ell - i - 1$ steps, completing the induction.

---

[3] Since the induction ends in $\ell$ steps, a sufficiently large $n_i$ can always be realized by a sufficiently large $n$.

