# OpenReview forum: "Diffusion Language Models are Provably Optimal Parallel Samplers"
_ICLR.cc/2026/Conference — ICLR 2026 Poster_

### Official Review · Reviewer_9o2i · 2025-10-28

**Soundness:** 3
**Presentation:** 2
**Contribution:** 3
**Rating:** 6
**Confidence:** 2

**Summary:**

The paper provides a theoretical justification on why Diffusion language models (DLMs) can simulate any parallel sampling algorithm using an optimal number of sequential steps. It makes use of the theory of complexity theory and boolean circuits. First, it proves that DLM with CoT can generate any distribution (on binary data) with the minimum possible sequential steps. Second, it shows that DLMs empowered with the remasking and revision abilities, can sample from any distribution with the minimum amount of memory. Finally, the authors characterize more precisely the simulation abilities of DLMs with/without remasking or/and revision, showing  that DLMs with remasking or revision are more expressive.

Overall, the paper is theoretical and contains no experiments.

**Strengths:**

The paper provides a theory of diffusion language models (DLMs) in the context of binary data and boolean circuits, precisely characterising DLMs expressivity and their ability to simulate parallel sampling algorithm with optimality characteristics. To the best of my knowledge, this is the first paper which builds up a theory for DLMs using such an angle.

The paper contains the key results and all the proofs.

**Weaknesses:**

1. The authors could steer the readers better at the beginning of the paper on how and why the boolean circuits theory is relevant for understanding DLMs. While the authors provide a presentation & background on boolean circuits, what is missing is the step on how such theory could generalize our understanding of DLMs on arbitrary vocabularies.

2. In boolean circuits presentation, the authors introduce class TC with MAJORITY operation, but such operation is never used (nor this class). It seems to be coming out of nowhere, especially for readers who are not familiar with boolean circuits. I think authors could do a better job to guide the reader on why such complexity classes are introduced in the way they are introduced. Moreover, the authors say "By definition, it is known that for all i \in N, ...", but no reference to such a statement is provided.

3. While I understand that the theory the authors propose is about understanding the expressivity and optimality of "ideal DLMs", such theory does not tell us much about what abilities we would have once we train the DLMs. I think the authors should at least include a discussion about that in their paper.

4. The authors introduce CoT as essentially some intermediate sample steps which DLMs could perform, but CoT is more than that -- it is highly structured prompt of very specific type. The theory proposed in the paper does not make this distinction. I think therefore term CoT might not be correct as it is related to a broader literature. Could authors please explain how CoT they describe connects to CoT which is generally used in the LLM litterature?

**Questions:**

Please see the "Weaknesses" section.

1. Can you provide a more clear connection on the relevance of boolean circuits and your theory angle to a broader understanding of DLMs?

2. Can you explain the complexity classes for boolean circuits and why they are introduced the way they are introduced ? Adding some references would be helpful.

3. How does the proposed theory relate to trained DLMs (which can have some approximation errors?)

4. How does your notion of CoT related to a broadly understood notion of CoT?

---

> ### Author Response · Authors · 2025-11-27
> **Rebuttal**
>
> We thank the reviewer for their positive and careful review and valuable feedback. In the following we address the reviewer's questions:
>
> > The authors could steer the readers better at the beginning of the paper on how and why the boolean circuits theory is relevant for understanding DLMs.
>
> We add a paragraph in the updated version to clarify the usage of Boolean circuits in this paper.
>
> > I think authors could do a better job to guide the reader on why such complexity classes are introduced in the way they are introduced.
>
> We update the paragraph on circuit complexity classes by introducing the three classes in a smoother succession. We add a reference to their relations as well. You are correct that in the original version of the paper $\mathsf{TC}$ class never appears. However, we keep this for now as it comes up in the rebuttal. We also add a paragraph in Section 2.2 explaining the usage of circuit models in this paper.
>
> > While I understand that the theory the authors propose is about understanding the expressivity and optimality of "ideal DLMs", such theory does not tell us much about what abilities we would have once we train the DLMs. I think the authors should at least include a discussion about that in their paper.
>
> It is indeed an interesting future direction to explore whether standard training techniques could find highly parallelizable solutions. On the empirical side, a lot of diffusion language models have shown extraordinary inference speed, especially on tasks like coding that can benefit greatly from parallelized decoding ([1,2,3]). Our results agree with these empirical observations and provide theoretical justifications. Besides, there have been growing interests in augmenting training data by adding revisions in the forward process to teach the language model to revise unmasked tokens while decoding ([3]). We believe that designing principled ways to design the forward process and collect training data that contains revision/remasking can force the model to utilize revision/remasking effectively, and this is a very important direction. We have incorporated this discussion into Section 4 of the updated paper.
>
> > The authors introduce CoT as essentially some intermediate sample steps which DLMs could perform, but CoT is more than that -- it is highly structured prompt of very specific type. The theory proposed in the paper does not make this distinction. I think therefore term CoT might not be correct as it is related to a broader literature. Could authors please explain how CoT they describe connects to CoT which is generally used in the LLM litterature?
>
> We agree that in practice the technique of chain of thought has various forms, including using structured prompts to elicit longer output sequences and ultimately lead to better performance. Theoretically speaking, all these techniques can be formalized as eliciting intermediate output to enhance the accuracy of the final output. Hence, in the theoretical study of the expressive power of language models, we formalize CoT as allowing for intermediate outputs before the final output. It is well established in literature to use intermediate sample steps as a theoretical model to study the expressive power of chain of thoughts ([4]), and our paper adopts this formalism as well.
>
> [1] Zhihui Xie, Jiacheng Ye, Lin Zheng, Jiahui Gao, Jingwei Dong, Zirui Wu, Xueliang Zhao, Shansan Gong, Xin Jiang, Zhenguo Li, and Lingpeng Kong. Dream-coder 7b: An open diffusion language model for code, 2025. URL https://arxiv.org/abs/2509.01142.
>
> [2] Samar Khanna, Siddhant Kharbanda, Shufan Li, Harshit Varma, Eric Wang, Sawyer Birnbaum, Ziyang Luo, Yanis Miraoui, Akash Palrecha, Stefano Ermon, et al. Mercury: Ultra-fast language models based on diffusion. arXiv preprint arXiv:2506.17298, 2025.
>
> [3] Yuxuan Song, Zheng Zhang, Cheng Luo, Pengyang Gao, Fan Xia, Hao Luo, Zheng Li, Yuehang Yang, Hongli Yu, Xingwei Qu, et al. Seed diffusion: A large-scale diffusion language model with high-speed inference. arXiv preprint arXiv:2508.02193, 2025.
>
> [4] Zhiyuan Li, Hong Liu, Denny Zhou, and Tengyu Ma. Chain of thought empowers transformers to solve inherently serial problems. In The Twelfth International Conference on Learning Represen- tations, 2024.

---

### Official Review · Reviewer_kw6r · 2025-10-29

**Soundness:** 3
**Presentation:** 3
**Contribution:** 3
**Rating:** 4
**Confidence:** 3

**Summary:**

This paper studies diffusion language models (DLMs) which recently appeared as an alternative to autoregressive models for their efficiency at inference time (possibility to generate tokens in parallel). It provides theoretical results on the benefits of DLMs for parallel sampling from the perspetive of implementing boolean circuits. This allow to quantify the efficiency of DLMs in terms of circuits classes, depth and bits to implement them. The main results show that for any circuit of depth $d$, there exists a DLM with chain-of-thought (CoT) that can implement it with $d$ decoding steps. This comes with additional large intermediate footprints. Further allowing DLMs to use remasking and revision extends the efficiency and expressivity of DLMs since they become strictly more expressive than DLMs without revision or remasking.

**Strengths:**

- The paper is well-written and enough context is given for the reader on the notions used in the paper
- The subject is interesting since diffusion LMs are more and more used so a better theoretical understanding is of importance
- The circuit formulation is elegant and the proofs seem sound although I did not check all of the details in appendix
- The theoretical claims are of great interest and the methodology of comparing DLMs with and without remasking and revision is well conducted
- The proofs techniques could be of independent interests to study complex models such as neural networks and language models.

**Weaknesses:**

I list below what I believe are weaknesses but I would be happy to be corrected if I misunderstood some parts.
- The connection to distribution sampling should be made more explicit
- The succession of theoretical results without much discussion on their insights for diffusion LMs hinders the contributions
- Thm 3.1 is one of the main result but only an existence results. As such, it does not ensure that any DLM can simulate any distribution sampling efficiencly nor does it provide guarantees on the existing DLM that can. As such, I find the presentation of this results in abstract ("We prove that enabling remasking (converting
unmasked tokens to masks or revision (converting unmasked tokens to other unmasked tokens) together with CoT further allows DLMs to simulate any parallel sampling algorithm with optimal space complexity") and the rest of the paper misleading. Could the authors elaborate on that (see questions)?
- Those two weaknessess explained above make unclear the benefits of the current work. While showing the benefits of remasking and revision is interesting, the connection to distribution sampling is unclear and the connection to practical benefits observed in practice is unclear. I believe the paper would be improved with more discussion on that.

Overall, my main issue with the current work is the connection to distribution sampling that is not clear enough and as such, the benefits of the results to explain the current (and potential future) success of DLMs in terms of efficiency and expressivity is unclear beyond the boolean circuit framework. I believe that better connecting that to distribution sampling and practical use of DLMs (still with theoretical results) would improve the current work. This is why I currently lean towards rejecting the paper but a convincing connection would make me lean towards accepting it.

**Questions:**

- Thm 3.1 is one of the main results and it is an existence result but does not ensure that a given DLM can achieve this low number of decoding steps. As such, its benefits for practial DLMs is unclear. Could the authors elaborate on that point?
- Why not considering prompts in Section 4? Since it echoes results of Section 3, it would be interesting to have similar settings. Could the authors elaborate on that?
- Obtaining meaningful theoretical results often require simplifying assumptions. In line 131, the authors assume that the length with CoT is $L$ while in practice $L$ is larger than q|CoT|o. How would the results change if considering a bigger sequence length $L$?
- Remasking is assumed to be modeled by a random function $G$ ni line 138. What is it in practice? If different than random, how does it impact the theoretical results?

*Typos*
- line 176: "the" should be removed
- line 176: $(x, R)$ should be $(\chi, R)$
- line 200: $\mathbb{Z}^+$ is simply the set of integers $\mathbb{N}$.
- line 202: "unmasked" --> "unmask"

---

> ### Author Response · Authors · 2025-11-27
> **Rebuttal**
>
> We thank the reviewer for their careful review and valuable feedback. In the following we address the reviewer's questions:
>
> > The connection to distribution sampling should be made more explicit
>
> In the updated version, we add a section that explicitly states the relationship of circuit models and distribution sampling, as well as modelling the DLM itself. Since most modern computers carry out computations using pre-specified operations, it is natural to use circuits to model the distributions that can be sampled by computers, as well as the computation in DLMs.
>
> > The succession of theoretical results without much discussion on their insights for diffusion LMs hinders the contributions
>
> In the updated version, we add more discussion on implications on computational costs, motivations for studying the separation results, and implications on revision/remasking to better connect the theoretical contents.
>
> > Thm 3.1 [...] is an existence result but does not ensure that a given DLM can achieve this low number of decoding steps. As such, its benefits for practical DLMs is unclear.
>
> This paper focuses on the expressive power of diffusion language models, similar to a lot of prior works ([1,2]). It is indeed an interesting future direction to explore whether standard training techniques could find highly parallelizable solutions. On the empirical side, a lot of diffusion language models have shown extraordinary inference speed, especially on tasks like coding that can benefit greatly from parallelized decoding ([3,4,5]). Our results agree with these empirical observations and provide theoretical justifications. Besides, there have been growing interests in augmenting training data by adding revisions in the forward process to teach the language model to revise unmasked tokens while decoding ([5]). We believe that designing principled ways to design the forward process and collect training data that contains revision/remasking can force the model to utilize revision/remasking effectively, and this is a very important direction. We have incorporated this discussion into Section 4 of the updated paper.
>
> > Why not considering prompts in Section 4? Since it echoes results of Section 3, it would be interesting to have similar settings. Could the authors elaborate on that?
>
> The main purpose of Section 4 is to show that there is a strict separation between DLM with and without remasking/revision. We use the simplest distribution that we could think of to show the separation for clarity. In contrast, in Section 3 we want to show the efficiency of DLMs in the most general case, so working with prompts makes the arguments stronger. For Section 4, based on the proposed distribution $\mathcal{D}_n^\oplus$, we could easily construct conditional distributions (output distribution conditioned on prompts) to show the same separation. For example, we can construct a distribution, such that for some prompt, the output distribution is $\mathcal{D}_n^\oplus$, and for the other prompts, the output distributions are easy to simulate by $\mathsf{AC}^0$ circuits.
>
> > Obtaining meaningful theoretical results often require simplifying assumptions. In line 131, the authors assume that the length with CoT is $L$ while in practice $L$ is larger than q|CoT|o. How would the results change if considering a bigger sequence length $L$?
>
> We choose the current formulation purely for notation simplicity, the results would not change significantly if we had a bigger sequence length. For results in Section 3, we just need to output an extra $\texttt{EOS}$ token following $o$ to indicate the position of output. For the results in Section 4, since we choose the distribution for separation results according to window length, the result does not change at all.
>
> > Typos
>
> We have fixed all the typos in the updated version.
>
>
> [1] Zhiyuan Li, Hong Liu, Denny Zhou, and Tengyu Ma. Chain of thought empowers transformers to solve inherently serial problems. In The Twelfth International Conference on Learning Representations, 2024.
>
> [2] William Merrill and Ashish Sabharwal. The parallelism tradeoff: Limitations of log-precision transformers. Transactions of the Association for Computational Linguistics, 11:531–545, 2023.
>
> [3] Zhihui Xie, Jiacheng Ye, Lin Zheng, Jiahui Gao, Jingwei Dong, Zirui Wu, Xueliang Zhao, Shansan Gong, Xin Jiang, Zhenguo Li, and Lingpeng Kong. Dream-coder 7b: An open diffusion language model for code, 2025. URL https://arxiv.org/abs/2509.01142.
>
> [4] Samar Khanna, Siddhant Kharbanda, Shufan Li, Harshit Varma, Eric Wang, Sawyer Birnbaum, Ziyang Luo, Yanis Miraoui, Akash Palrecha, Stefano Ermon, et al. Mercury: Ultra-fast language models based on diffusion. arXiv preprint arXiv:2506.17298, 2025.
>
> [5] Yuxuan Song, Zheng Zhang, Cheng Luo, Pengyang Gao, Fan Xia, Hao Luo, Zheng Li, Yuehang Yang, Hongli Yu, Xingwei Qu, et al. Seed diffusion: A large-scale diffusion language model with high-speed inference. arXiv preprint arXiv:2508.02193, 2025.

---

> > ### Comment · Reviewer_kw6r · 2025-11-27
> >
> > I thank the authors for addressing my concerns. Although the writing still hinders the contributions, the additional explanations and clarification regarding the distribution sampling improves the current submission. I update my score accordingly.

---

> > ### Comment · Reviewer_kw6r · 2025-11-27
> >
> > I thank the authors for addressing my concerns. Although the writing still hinders the contributions, the additional explanations and clarification regarding the distribution sampling improves the current submission. I update my score accordingly.

---

### Official Review · Reviewer_qnmR · 2025-10-30

**Soundness:** 3
**Presentation:** 2
**Contribution:** 3
**Rating:** 8
**Confidence:** 2

**Summary:**

In this paper, the authors derive some theoretical results on Diffusion Language Models.
In particular, they  show that random functions realized by a circuit with depth $d$ can be described by a Diffusion Language Model with $d$ decoding steps (when augmented with CoT).
Interestingly they show that the length of the sequence to be generated with a (masked) diffusion model to generate a circuit of size $N$ is $N$ while in the case of a forward process with non-masked perturbation or in the case where remasking is used at inference time they can generate the circuit with a sequence length of the width of the circuit.
Finally, the authors show that if we restrict the building block circuits for the Diffusion Language Model to be in a certain complexity class then  there is exists an example for which Diffusion Language Models with forward perturbation or remasking can achieve the reconstruction of the examplar circuit in $O(1)$ steps and such a bound is not attainable without the ability to remask tokens.

**Strengths:**

For me the importance of the results is two fold:

* First, they show the impact of remasking (or using a non-masking forward process) by showing that not only Diffusion Language Models can simulate any circuit but also by showing that the sequence length necessary to generate such circuits is limited by the width of the circuit in the case of Diffusion Language Model.

* Second they highlight an example showing the superiority of Diffusion Language Models with remasking (or using a non-masking forward process) by showing that such Diffusion Language Models can achieve the reconstruction of an examplar circuit in $O(1)$ steps and such a bound is not attainable without the ability to remask tokens.

**Weaknesses:**

I do not have a lot of complains about the paper.
I will highlight that I am not an expert on circuits so I did find some parts of the paper hard to follow.


* I would suggest to clarify the writing especially Section 4. Indeed, in this Section while I understood the results and their consequences I was unable to follow the logical structure. (Again it might be acceptable for experts in circuits but I could not follow).

* Importance of the results: I am not fully convinced that being able to generate circuit is indicative of performance of Diffusion Language Models. In particular, are there real-word scenarios which are well described by the circuit framework the authors describe (sudoku, puzzle?). It is unclear to me that in the case of _text_ this analysis remains relevant. I would appreciate a discussion about this potential limitation.

* All the results rely on CoT, it would be more impactful to understand what would happen without CoT assumptions.

* All the proofs and the main points of the paper rely on the fact that we can perform parallel sampling operations within the framework of Diffusion Language Model. One big omission in my opinion is that speculative decoding/sampling are powerful methods for AR models which unlock parallel sampling for AR models. I think those models and the consequences of the analysis presented in the current paper and its extension to spec decoding is key to understand what are the key ingredients of the proof and what is truly specific to DLM

**Questions:**

* The authors state that "More recently, DLMs with revision (Song et al., 2025) are introduced and exhibit remarkable capability."
It is not really clear to me, how the "revision" technique differs from a uniform discrete diffusion. In that case, that would mean that we leave the masking world and that strategies such as remasking are not needed. I am wondering if the current framework also allows the authors to understand this scenario?

* Related to this question, I am wondering if the strategy of the authors could be used to understand the benefits of current continuous augmentation strategies such as CCDD [1] .

* What would be the step to extend the results beyond binary vocabulary? What are the blockers?

* Typo line 176 (notation for the input)


[1] Zhou et al., (2025) -- Coevolutionary Continuous Discrete Diffusion: Make Your Diffusion Language Model a Latent Reasoner

---

> ### Author Response · Authors · 2025-11-27
> **Rebuttal**
>
> We thank the reviewer for their positive and careful review and valuable feedback. In the following we address the reviewer's questions:
>
> > I would suggest to clarify the writing especially Section 4.
>
> We have rewritten the proof of Theorem 4.5 to a logic structure that is easier to follow.
>
> > In particular, are there real-world scenarios that are well described by the circuit framework the authors describe (sudoku, puzzle?). It is unclear to me that in the case of text, this analysis remains relevant. I would appreciate a discussion about this potential limitation.
>
> Circuits have been widely used as a model to capture the computation time needed when we have access to parallel computers. For any task that a computer carries out, such as solving Sudoku and puzzle, everything is encoded in a binary form, and the computations of an algorithm carried out by modern computers are well modeled by circuit models. Hence we believe that the circuit framework is realistic. As a matter of fact, there is a long line of work using circuit complexity to analyze the expressiveness of architectures that are widely used to process texts ([1,2]). We incorporate this into the updated version in Section 2.2.
>
> > All the results rely on CoT, it would be more impactful to understand what would happen without CoT assumptions.
>
> We would like to point out that studying the expressive power of DLMs without CoT is unrealistic. In practice, DLM generates the output sequence with a pre-specified length $L$, and determines the actual output by truncating the sequence at the first special token $\texttt{EOS}$ generated. Hence in principle we do not have the power to control that intermediate output tokens do not exist. That is to say, assuming that CoTs exist in general best represents that real-world scenario.
>
> > I think those models and the consequences of the analysis presented in the current paper and its extension to spec decoding is key to understand what are the key ingredients of the proof and what is truly specific to DLM.
>
> We totally agree that it is very important to compare DLM with other architectures that allow for parallel decoding, and this indeed is a very interesting direction to investigate. The success of speculative decoding crucially relies on the capability difference between the draft model and the target model. While this paper focuses on expressive power only, we believe capturing capability needs extra formalism and is certainly an interesting future direction.
>
> > It is not really clear to me, how the "revision" technique differs from a uniform discrete diffusion. [...] I am wondering if the current framework also allows the authors to understand this scenario?
>
> In Seed Diffusion (Song et al., 2025), an edit-based forward process is introduced. Rather than uniformly randomly masking the sequence, they also allow the sequence to change according to predefined operations, therefore allowing tokens to change in the forward process. In this way, the model can be taught to revise tokens instead of just unmasking. In our framework, the DLM with revision indeed captures this scenario.
>
> > Related to this question, I am wondering if the strategy of the authors could be used to understand the benefits of current continuous augmentation strategies such as CCDD.
>
> Yes, we think our framework can be used to analyze CCDD. We need to discretize the continuous diffusion part of CCDD to include it in the circuit framework, and it is indeed an interesting future direction.
>
> > What would be the step to extend the results beyond binary vocabulary? What are the blockers?
>
> As explained in footnote 1, we use binary vocabulary in this paper to simplify the exposition of the results. There is no blocker in principle to extend the result beyond a binary vocabulary. We just need to change the encoding of vocabulary and use more bits to represent a token.
>
> > Typo line 176
>
> We fixed the typo in the updated version.
>
> [1] Zhiyuan Li, Hong Liu, Denny Zhou, and Tengyu Ma. Chain of thought empowers transformers to solve inherently serial problems. In The Twelfth International Conference on Learning Representations, 2024.
>
> [2] William Merrill and Ashish Sabharwal. The parallelism tradeoff: Limitations of log-precision transformers. Transactions of the Association for Computational Linguistics, 11:531–545, 2023.

---

### Official Review · Reviewer_vuFd · 2025-10-31

**Soundness:** 3
**Presentation:** 3
**Contribution:** 2
**Rating:** 4
**Confidence:** 3

**Summary:**

This paper provides a theoretical analysis of Diffusion Language Models (DLMs) by leveraging the framework of  Boolean circuit complexity. DLMs have become very popular recently as they allow faster inference than Autoregressive (AR) models via parallel decoding. The manuscript represents parallel computation using circuit depth (time) and circuit width (space). They analyze DLMs with some (unusual) Chain-of-Thought (CoT), acting as a kind of scratchpad, and two increasingly popular inference-time mechanisms: remasking and updating (modifying already unmasked tokens).

The paper establishes the following results:
Theorem 3.1 shows that DLMs using sufficient CoT can simulate any depth-$d$ circuit in exactly $d$ rounds. This achieves the theoretical minimum for parallel computation, contrary to AR models which require steps proportional to circuit size.
Theorem 3.2 shows that by remasking and updating with CoT, DLMs can achieve optimal time ($O(d)$ steps) and optimal space (memory proportional to circuit width $w$).
In Section 4, the paper shows that standard DLMs with components restricted to the $AC^0$ complexity class cannot sample the uniform parity distribution in constant time, whereas DLMs with updating/revision or remasking can.

**Strengths:**

Originality: The paper is original as it is (to the best of my knowledge) the first paper that analyzes the parallel sampling capabilities of DLMs through circuit complexity ideas, so it is an original combination of ideas from theoretical CS and generative modeling.

Quality. The main results are rigorous and the proofs are constructive. I am not an expert on this type of results but could follow (most of) them. Theorem 3.1 provides an equivalence between circuit depth and decoding rounds. The construction in Theorem 3.2 appears non-trivial, it achieves near-optimal space complexity using remasking/updating. The expressivity separation of Section 4 provides a concrete example demonstrating the limitations of standard DLMs.

Clarity. The paper is quite difficult to read but it is also because it brings two literatures together. It provides a  formalization of the DLM inference process (Algorithm 1) which defines quite clearly the roles of the predictor, the scheduler, CoT, remasking, and updating/revision. The mapping between these components and Boolean circuits is clarified in Section 2.2 and is key to the understanding of the theoretical analysis.

Significance. This provides positive results justifying the advantage of DLMs over AR models, as sequential steps scale with circuit size rather than depth. It also shows that heuristic techniques proposed to improve DLMs (updating/remasking) are indeed mechanisms necessary to achieve optimal space efficiency and expressivity.

**Weaknesses:**

My main concerns are how these theoretical results relate to practical DLM training and inference; i.e. the presented results are interesting but I am not convinced they are telling us much in particular about existing DLMs and whether there is a path to exploit these results to obtain better DLMs.

Expressivity and Learnability. The main results (Thms 3.1, 3.2) are existence proofs. While they show that an optimal predictor $p$ and scheduler $\mathcal{F}$  exist for specific circuit classes, they fall short of addressing whether these can be learned by standard architectures using standard ELBO objective. It would be beneficial to explain (empirically or theoretically) whether standard training technique can be learned via standard stochastic gradient techniques or whether these techniques tend to find less parallelizable solutions.

Computational Cost. Optimality is defined as minimizing the number of sequential rounds. This does not really take into account the actual computational cost or wall-clock latency of using predictor $p$ and scheduler $\mathcal{F}$ in each round. It would be useful to clarify that the results do not appear to guarantee lower overall FLOPs compared to AR models (correct me if i am wrong), only lower latency for an idealized parallel hardware.

Complexity of the unmasking schedule. The optimality results appear to rely on an unput-dependent, and dynamic unmasking schedule $\mathcal{F}$ that precisely identifies which tokens (circuit nodes) to compute next. This is much more complex that what is typically used in practice. It would be beneficial to discuss the schedule$\mathcal{F}$ . How robust are the results if you were to modify the $\mathcal{F}$, would one be far off the optimal? how would we learn it practically?

$AC^0$ Constraints. The  results of Section 4 appear to rely heavily on the assumption that the predictor and scheduler are constrained to $AC^0$. However, $AC^0$ is a weak complexity class. How would these results translate to practical scenarios where one  has access to much more powerful predictors, e.g., Transformers. It would be useful to clarify the implications of the  $AC^0$ constraint. Would the  architectural separation outlined in section 4 still hold for much more powerful predictors?

CoT. I find the reliance on CoT quite disturbing... In particular because the CoT considered in the theoretical arguments appears to differs very significantly from practical CoT, i.e. a chain of reasoning steps. I think this requires very substantial clarification.

**Questions:**

I think it is accepted that transformers are more powerful than $AC^0$. What happens to Theorem 4.5 in this case? If the separation does not hold anymore, what does this imply about the necessity of updating/revision for expressivity in practice?

Is there any evidence, whether theoretical or empirical, that standard diffusion training objectives encourage the model to discover the clever time/space-optimal strategies  outlined in  Theorems 3.1 and 3.2? If it is not the case, can you think of any objective that would explicitly force he model to utilize revision/remasking effectively for few-round decoding?

How robust is the $O(d)$ round complexity to the optimality of the scheduler $\mathcal{F}$? Do existing suboptimal (unmask random tokens) or adaptive schedules (i.e. greedy decoding) significantly increase the required rounds?

Beyond parity, can you think of other types of distributions exhibit hardness for standard DLMs but become tractable with updating/revision?

---

> ### Author Response · Authors · 2025-11-27
> **Rebuttal (Part I)**
>
> We thank the reviewer for their careful review and valuable feedback. The reviewer raises concerns about the gap between theory and practice. In the following we address the reviewer's questions:
>
> > The main results are existence proofs. [...] explain whether standard training techniques tend to find parallelizable solutions.
>
> This paper focuses on the expressive power of diffusion language models, similar to a lot of prior works ([1,2]). It is indeed an interesting future direction to explore whether standard training techniques, such as stochastic gradient descent, could find highly parallelizable solutions. On the empirical side, a lot of diffusion language models have shown extraordinary inference speed, especially on tasks like coding that can benefit greatly from parallelized decoding ([3,4,5]), so this suggests certain parallel solutions can be learned in practice.
>
> > This does not really take into account the actual computational cost or wall-clock latency. [...] It would be useful to clarify that the results do not appear to guarantee lower overall FLOPs compared to AR models, only lower latency for an idealized parallel hardware.
>
> You are right that our results do not imply that the computation cost or wall-clock latency for DLM would be better than autoregressive models. We clarify this point in the updated version at the end of Section 3.1. As a concrete example, we compare the computation cost and wall-clock latency in one decoding step between an autoregressive Transformer with KV caching and a DLM implemented by a Transformer of the same size. In every Transformer block, an autoregressive model only needs to compute the attention output for one position, whereas a DLM needs to compute the attention output for all positions. Hence a DLM needs $O(L)$ (the number of tokens) times more FLOPs than an autoregressive model. The computation of the $L$ tokens is parallelizable for the DLM, so if the hardware supports enough degree of parallelism, they have the same wall-clock latency. However if the hardware does not support enough parallel computing, the wall-clock latency for the DLM can be slower.
>
> > The optimality results appear to rely on an input-dependent and dynamic unmasking schedule. [...] would one be far off the optimal? how would we learn it practically?
>
> We would like to point out that a lot of $\mathcal{F}$ used in practice also depend on inputs and induce a dynamic unmasking schedule, and the design of $\mathcal{F}$ has become an important topic. As an example, we may choose the token with the highest confidence to decode, and this has been proven to greatly improve DLM's capability. One may refer to [6] for more details and decoding strategies. Even more, people recently started parameterizing $\mathcal{F}$ as neural networks to further enhance model capability ([13]). Besides, the expressiveness of $\mathcal{F}$ is not unbounded. We constrain it to be constant-depth circuit with AND, OR and NOT. Furthermore, the $\mathcal{F}$ circuits constructed in the proofs are all within $\mathsf{NC}^0$, one of the least expressive circuit classes one can think of. Hence we believe that our formulation reflects the reality, and justify that a flexible decoding strategy can make DLMs more powerful.

---

> > ### Author Response · Authors · 2025-11-27
> > **Rebuttal (Part II)**
> >
> > > $\mathsf{AC}^0$ Constraints. [...] Would the architectural separation outlined in section 4 still hold for much more powerful predictors?
> > > I think it is accepted that transformers are more powerful than $\mathsf{AC}^0$. What happens to Theorem 4.5 in this case? If the separation does not hold anymore, what does this imply about the necessity of updating/revision for expressivity in practice?
> >
> > As explained in the paper, prior works have shown that a constant-precision Transformer can be simulated by an $\mathsf{AC}^0$ circuit. (add a citation?) We believe that constant precision is a reasonable assumption for modern large language models, as people are often using low-precision floating numbers for inference in larger models. Concretely, widely-used open-source model families like Mistral/Mixtral, Qwen, Deepseek, LLaMA, etc. are natively 16-bit on most of their models across different sizes. Furthermore, there is a long line of work showing that quantization to 4-8 bits at inference time is widely reliable ([7,8,9]). Even more, in BitNet ([10]), it is shown that a sub-2-bit model could match the performance of LLaMA 2 70B model with the same parameter count. If we postulate that Transformers are more expressive than $\mathsf{AC}^0$, we would need a new distribution for the separation result. For example, if we postulate that Transformers are log-precision and hence can be simulated by $\mathsf{TC}^0$ circuits ([1]). It is promising to use the uniform distribution over sequences of group elements from the permutation group $S_5$ with product identity as separation. Generating such a distribution with DLM with revision is similar to that of parity, and it is known that determining whether a sequence's product is identity in $S_5$ is $\mathsf{NC}^1$-complete ([11]). Under the assumption that $\mathsf{TC}^0\not\subseteq\mathsf{NC}^1$, a standard circuit complexity assumption, we could get a separation using similar proof techniques.
> >
> > > CoT considered in the theoretical arguments appears to differs very significantly from practical CoT, i.e. a chain of reasoning steps. I think this requires very substantial clarification.
> >
> > We agree that in practice the technique of chain of thought has various forms. Theoretically speaking, all these techniques can be formalized as eliciting intermediate outputs to enhance the accuracy of the final output. Hence, in the theoretical study of the expressive power of language models, we formalize CoT as allowing for intermediate outputs before the final output, while abstracting away the technique that elicits such behavior. It is well established in literature to use intermediate sample steps as a theoretical model to study the expressive power of chain of thoughts ([1]), and our paper adopts this formalism as well.
> >
> > > Is there any evidence, whether theoretical or empirical, that standard diffusion training objectives encourage the model to discover the clever time/space-optimal strategies outlined in Theorems 3.1 and 3.2? If it is not the case, can you think of any objective that would explicitly force he model to utilize revision/remasking effectively for few-round decoding?
> >
> > At this moment, there is no evidence that the standard training objective encourages the model to discover the exact constructions outlined in the proof. However, there have been growing interests in augmenting training data by adding revisions in the forward process to teach the language model to revise unmasked tokens while decoding ([5]). We believe that designing principled ways to design the forward process and collect training data that contains revision/remasking can force the model to utilize revision/remasking effectively, and this is a very important direction. We have incorporated this discussion into Section 4 of the updated paper.
> >
> > > How robust is the $O(d)$ round complexity to the optimality of the scheduler $\mathcal{F}$? Do existing suboptimal (unmask random tokens) or adaptive schedules (i.e. greedy decoding) significantly increase the required rounds?
> >
> > With a random unmasking strategy, we do not expect that sampling any distribution is possible with constant-depth predictors. This is because even if a token requires $O(d)$ steps to generate, there is a possibility that we require this position to be decoded in the first step. This is consistent with the fact that random decoding usually yields suboptimal performance ([6]). It is easy to convert the current construction into a proof with greedy decoding. We just need to add another layer of circuit, which assigns very high confidence to the tokens output by the predictor $p$ at position by the scheduler $\mathcal{F}$ as to be decoded, and assigns very low confidence to other tokens. Hence, if we use greedy decoding, we could still achieve $O(d)$ rounds of decoding. Of course, there are more decoding strategies. However, we believe that our framework is realistic and general enough to study most practical design choices.

---

> ### Author Response · Authors · 2025-11-27
> **Rebuttal (Part III)**
>
> > Beyond parity, can you think of other types of distributions exhibit hardness for standard DLMs but become tractable with updating/revision?
>
> Let us first summarize the intuition on why parity can be used to show such separation and then informally describe how we can use more distributions to show separation.
>
> Recall that in order to sample a uniform distribution over zero parity, we first sample $y_1,\dotsc,y_{n-1}$ uniformly at random, and then set $x_i = y_{i} \oplus y_{i-1}$, where $y_0 = 0$ and $y_n = 0$. It is hard to sample $x_n$ conditioned on $x_1,\cdots,x_{n-1}$ because $x_n$ depends globally on all $x_1,\cdots,x_{n-1}$. The calculation cannot be done with constant in-degree gates. It is easy to sample $x_1,\cdots,x_n$ from $y_1,\cdots,y_n$ because $y_i$ depends only locally on $x_i,x_{i-1}$. Any distribution that satisfies these two properties on variable dependencies can be used to show the separation. One example is the uniform distribution over sequence whose sum equals zero modulo $m$ where $m$ is an integer bigger than two.
>
>
> [1] Zhiyuan Li, Hong Liu, Denny Zhou, and Tengyu Ma. Chain of thought empowers transformers to solve inherently serial problems. In The Twelfth International Conference on Learning Represen- tations, 2024.
>
> [2] William Merrill and Ashish Sabharwal. The parallelism tradeoff: Limitations of log-precision transformers. Transactions of the Association for Computational Linguistics, 11:531–545, 2023.
>
> [3] Zhihui Xie, Jiacheng Ye, Lin Zheng, Jiahui Gao, Jingwei Dong, Zirui Wu, Xueliang Zhao, Shansan Gong, Xin Jiang, Zhenguo Li, and Lingpeng Kong. Dream-coder 7b: An open diffusion language model for code, 2025. URL https://arxiv.org/abs/2509.01142.
>
> [4] Samar Khanna, Siddhant Kharbanda, Shufan Li, Harshit Varma, Eric Wang, Sawyer Birnbaum, Ziyang Luo, Yanis Miraoui, Akash Palrecha, Stefano Ermon, et al. Mercury: Ultra-fast language models based on diffusion. arXiv preprint arXiv:2506.17298, 2025.
>
> [5] Yuxuan Song, Zheng Zhang, Cheng Luo, Pengyang Gao, Fan Xia, Hao Luo, Zheng Li, Yuehang Yang, Hongli Yu, Xingwei Qu, et al. Seed diffusion: A large-scale diffusion language model with high-speed inference. arXiv preprint arXiv:2508.02193, 2025.
>
> [6] Jaeyeon Kim, Kulin Shah, Vasilis Kontonis, Sham Kakade, and Sitan Chen. Train for the worst, plan for the best: Understanding token ordering in masked diffusions. arXiv preprint arXiv:2502.06768, 2025.
>
> [7] Dettmers, T., Lewis, M., Belkada, Y., & Zettlemoyer, L. LLM.int8(): 8-bit matrix multiplication for transformers at scale. In Proceedings of the 36th Conference on Neural Information Processing Systems, 2022.
>
> [8] Xiao, G., Lin, J., Seznec, M., Wu, H., Demouth, J., & Han, S. (2023). SmoothQuant: Accurate and efficient post-training quantization for large language models. In Proceedings of the 40th International Conference on Machine Learning, 2023.
>
> [9] Frantar, E., Ashkboos, S., Hoefler, T., & Alistarh, D.. GPTQ: Accurate post-training quantization for generative pre-trained transformers. In The Eleventh International Conference on Learning Representations, 2023.
>
> [10] Ma, S., Wang, H., Ma, L., Wang, L., Wang, W., Huang, S., Dong, L., Wang, R., Xue, J., & Wei, F. The era of 1-bit LLMs: All large language models are in 1.58 bits. 2024. URL https://arxiv.org/abs/2402.17764.
>
> [11] David A. Barrington. Bounded-width polynomial-size branching programs recognize exactly those languages in nc. pp. 1–5, 1986.
>
> [12] Shen Nie, Fengqi Zhu, Zebin You, Xiaolu Zhang, Jingyang Ou, Jun Hu, Jun Zhou, Yankai Lin, Ji-Rong Wen, and Chongxuan Li. Large language diffusion models. arXiv preprint arXiv:2502.09992, 2025.
>
> [13] Huang, Z., Wang, Y., Chen, Z., & Qi, G. Don’t settle too early: Self-reflective remasking for diffusion language models. arXiv. 2025. https://arxiv.org/abs/2509.23653

---

### Author Response · Authors · 2025-12-03

We want to thank you for reviewing our paper. We know that the demands on ACs are high given the changes ICLR had to implement due to leaks, and we appreciate your time. To make it very easy for you, we summarize the state of the reviews and rebuttals:

Reviewer vuFd (score 4) compliments the originality of this paper and the justifications for DLM practices, and the main concern is that the theory is not well-connected to practice. In particular, it is not clear whether practical DLMs can discover the constructions in this paper, and the theoretical formulation of DLMs does not faithfully reflect the practice. To address the first problem, we argue that studying the expressive power of neural networks is a well-established means to study the capability potential of architectures by citing high-profile papers on the expressive power of Transformers. We also demonstrate that standard training techniques on DLMs can discover highly parallelized solutions by citing well-known empirical works in this field. Regarding the gap between theoretical formulation and practical implementation, including the formulation of CoT, the $\mathcal{F}$ function, expressive power of Transformers, and computation costs, we address the concerns by providing more comprehensive discussions on variants of DLM implementations, as well as citing high-profile papers in DLM theory to justify our formulation.

Reviewer qnmR (score 8) gives a very positive review, suggests better writing, and raises concerns about the gap between theory and practice. We have revised the writing as suggested by the reviewer, and addressed the gap between theory and practice in the same way as that for Reviewer vuFd.

Reviewer kw6r (score 4) compliments the writing, problem formulation, being significantly relevant to practice, and novelty in proof techniques. This reviewer suggests a more explicit connection to distribution sampling, a better succession of results, and a better connection to practice. We added corresponding discussions as suggested by the reviewer in the revised PDF, and before reverting the score, this reviewer raised the score to 6 after reading the rebuttal.

Reviewer 9o2i (score 6) compliments the originality of the paper, and raises concerns about writing and the connection between theory and practice. We improve the writing as suggested by the reviewer and address the theory-practice connection problem, the same as that for Reviewer vuFd.

In summary, we have improved the writing of this paper so that it provides a better explanation of the theoretical formulation and a better exposition of the results. We address the reviewers' concerns about the theory and practice gap by citing well-established research on neural network expressivity and discussions on variants of DLM implementations.

---

### Meta-Review · Area_Chair_w2Lx · 2026-01-07

**Summary:**

This paper is a theoretical study of diffusion LLM (DLMs). It presents a new theoretical perspective based on Boolean circuit complexity framework to explain the potential advantages of DLMs or AR models. The authors show that DLMs with certain CoT augmentation can simulate any parallel sampling algorithm with an optimal number of sequential steps. The paper includes several theorems stating this result in different level of generality. In particular, it shows CoT, remarking, and revision are crucial for DLMs.

**Reviewer Concerns:**

The reviews for this work are mixed. First of all, all reviewers agree this work is original and novel. It is a unique combination of TCS and DLMs. The biggest concern is on the practical impacts of the theoretical results established in this paper. All the theorems are existence results. It is unclear how such results can be used in practice. Some assumptions made in this paper are not aligned with practice. The CoT augmentation is puzzling. Overall, there seems to be a gap between the theoretical results and the training and inference of DLMs in practice, but the theoretical results are solid.

**Reviewer Scores:**

Reviewer kw6r could have increased score slightly

---

### Decision · Program_Chairs · 2026-01-26

Accept (Poster)